

# The mean reticulocyte volume is a valuable index in early diagnosis of cancer-related anemia

Huijun Lin[1,*], Bicui Zhan[2,*], Xiaoyan Shi[3], Dujin Feng[1], Shuting Tao[1], Mingyi Wo[1], Xianming Fei[1], Weizhong Wang[1] and Yan Yu[4]

[1] Laboratory Medicine Center, Department of Clinical Laboratory, Zhejiang Provincial People's Hospital (Affiliated People's Hospital), Hangzhou Medical College, Hangzhou, Zhejiang, China
[2] Department of Clinical Laboratory, Hangzhou Hospital of Traditional Chinese Medicine, Hangzhou, Zhejiang, China
[3] Department of Clinical Laboratory, South Jinjiang Branch of Jinjiang Hospital, Jinjiang, Fujian, China
[4] Center for Rehabilitation Medicine, Department of Ophthalmology, Zhejiang Provincial People's Hospital (Affiliated People's Hospital), Hangzhou Medical College, Hangzhou, Zhejiang, China
* These authors contributed equally to this work.

Corresponding authors
Weizhong Wang, wwz1688@163.com
Yan Yu, yuyan0202@163.com

## ABSTRACT

**Background.** Cancer-related anemia (CRA) is a functional iron deficient anemia, and the early diagnosis will improve the prognosis of the patients. This prospective study aimed to investigate the utility of mean reticulocyte volume (MRV) in the early diagnosis of CRA.

**Methods.** A total of 284 first-diagnosed cancer patients were enrolled, and the subjects were assigned anemia and non-anemia groups by hemoglobin (Hb) concentrations. The mature RBC and reticulocyte indices were detected with BC-7500 blood analyzer, and the MRV, reticulocyte hemoglobin (RHE) content, and reticulocyte production index (RPI) were obtained. ROC curves were constructed in identifying anemia diagnosed by the combination of RHE and RPI. An adjusted multivariate analyse and quartiles were used to assess the associations of MRV with early CRA diagnosed by combining RBC indices (MCV, MCH and MCHC), respectively.

**Results.** No statistical differences were observed in MCV, RHE and MRV levels between anemia and non-anemia subjects ($p > 0.05$). MRV exhibited a complete or high correlation with the RHE levels ($r = 1.000$, $p < 0.001$), or MCV, MCH, and MCHC in anemia patients ($R$: 0.575–0.820, $p < 0.001$). ROC curves analyse indicated a highest area under curve of 0.829 (95% CI [0.762–0.895]) and 0.884 (95% CI [0.831–0.936]) for MRV in identifying anemia in male and female patients, respectively ($p < 0.001$). When the optimal cutoff values of MRV were set at 100.95 fl in males and 98.35 fl in females, the sensitivity and specificity were 1.00 and 0.68, and 1.00 and 0.73, respectively. The regression analyse showed that, when being as a categorical variable, MRV showed an odds ratio of 19.111 (95% CI [6.985–52.288]; $p < 0.001$) for the incidence of CRA. The incidence of overall anemia demonstrated a more significant decrease trend along with the increase of MRV quartiles (p-trend $< 0.001$).

**Conclusion.** This study revealed that the MRV can be used as a convenient and sensitive index in early diagnosis of cancer-related anemia, and decreased MRV level may be the powerful predictor of overt anemia in cancer patients.

# INTRODUCTION

Cancer is the prominent cause of mortality worldwide, and anemia occurs commonly as the manifestation of progressive cancer, which is called "cancer-related anemia (CRA) (*Madeddu et al., 2021*). The CRA is an important comorbidity at diagnosis of cancer (*Madeddu et al., 2018*). It has been reported that CRA occurs in more than 30% of cancer patients with solid cancer before any treatment (*Birgegård et al., 2005*). The associations of anemia with worse prognosis of cancer patients have been clearly established, including decreased survival and risk of death (*Madeddu et al., 2021*), indicating anemic cancer patients will have poor prognosis (*Badheeb et al., 2023*). Therefore, early identification and diagnosis of CRA will be of importance to improve the prognosis of cancer patients.

Up to now, the mechanisms of development and progression of CRA remain unclear, which is bad for its effective diagnosis and treatment (*Wang et al., 2021*). However, many reasons cause the occurrence of CRA, and the common ones are nutritional deficiency, hemolysis, blood loss, hematopoietic suppression by tumor cells (*Prabhash et al., 2011*). Whether the anemia is resulted from certain reason, the production, destruction, or loss of red blood cells can be tracked in the etiology of CRA (*Gilreath, Stenehjem & Rodgers, 2014*). The pathogenesis of CRA involves a variety of mechanisms caused by the cancer-related low-grade chronic inflammation, and the hematologic characteristics are similar to those described in anemia involving in other chronic inflammatory pathologies (*Natalucci et al., 2021*). The chronic inflammation can lead to functional iron deficiency and impaired erythropoiesis, and the subsequent secretion of various chemokines (*Madeddu et al., 2021*; *Adamson, 2008*). Increases in systemic inflammation finally cause anemia *via* several overlapping mechanisms: a negative impact of erythroid precursors, reduced production of erythropoietin, shortened red blood cell survival, and impaired iron metabolism in the reticuloendothelial system (*Adamson, 2008*; *Madeddu et al., 2018*; *Spivak, 2002*; *Faquin, Schneider & Goldberg, 1992*). Furthermore, previous studies have indicated that CRA is usually normochromic and normocytic anemia with a low reticulocyte count, normal serum iron concentrations (*Adamson, 2008*; *Spivak, 2005*; *Miller et al., 1990*; *Aapro et al., 2012*). Therefore, the abnormalities of some indicators reflecting the abovementioned three ways of the etiology of CRA would be closely linked with anemia, and can be used in prediction, early diagnosis, and disease assessment of CRA.

According to the definition of the World Health Organization, anemia occurs when hemoglobin (Hb) levels drop below 120 g/L for women and 130 g/L for men (*McLean et al., 2009*), but the diagnostic criteria for anemia will vary based on the differences of ethnicity and region. Although Hb level is the most important indicator in diagnosis of anemia, low Hb level just indicates an overt anemia, and its diagnostic sensitivity is insufficient for anemia, and can not identify and diagnose the early stage of anemia development. It has been clear that peripheral blood reticulocyte count is an important index in the diagnosis, classification, and monitoring of anemic patients (*Wollmann et*

*al., 2014*). Several reticulocyte maturity indices, such as immature reticulocyte fraction (IRF), reticulocyte production index (RPI), and reticulocyte hemoglobin (RHE) content, can be helpful in the diagnosis of pathologies, and in monitoring bone marrow recovery (*Wollmann et al., 2014*; *Cascio & De Loughery, 2017*; *Alzu'bi et al., 2023*; *Chang & Kass, 1997*). Thus, the measurements of non-tradictional reticulocyte indices will be of notable significance in the early diagnosis of CRA. *Xu et al. (2016)* has reported that reticulocyte volume is a good index to screen for hereditary spherocytosis, and *Nair et al. (2015)* also indicated its specific utility in differentiating peripheral blood spherocytes of hereditary spherocytosis from other causes. Moreover, other authors revealed that the abnormalities of iron deficiency were associated with decreased reticulocyte volume in dog (*Steinberg & Olver, 2005*), and different reticulocyte volume was observed in diabetes mellitus patients (*Bunyaratvej, Komindr & Wisedpanichkij, 2000*). However, rare reports were found about the utility of reticulocyte volume in CRA. This study was undertaken to study the utility of mean reticulocyte volume (MRV) in early identifying anemia from non-anemia in overall patients with variety of malignancies.

## MATERIALS AND METHODS

### Sample size

Before this study, the sample size was calculated as follows: $n = \frac{u^2 \sigma^2}{\delta^2}$ ($n$: total sample size; $\sigma$: total standard deviation; $\delta$: allowable error; $u$: U value under $a = 0.05$). In our preliminary study, the mean hemoglobin concentrations of anemic and non-anemic patients were 103 g/L and 139 g/L, respectively. When we set the allowable CV% as 2.0%, the allowable error ($\delta$) was calculated as 2.8 and 2.1 g/L by using abovementioned mean hemoglobin concentrations. According to the Chinese adult reference intervals for blood cell analysis (*Shang et al., 2013*), the total standard deviation ($\sigma$) is 9.2 and 8.7 g/L for males and females, respectively, and the mean $\sigma$ is 9.0 g/L. Therefore, we used the U value (1.96), $\delta$ and $\sigma$ to calculate the minimal sample size as 72 and 40 for anemic and non-anemic subjects, respectively, in this study. Moreover, we also set the percentage of missing cases as 20%, and the final sample size was 87 and 48.

### Study population

This is a cross-sectional study. A total of 284 consecutive patients with confirmed diagnosis of different solid malignancies were enrolled after applying the inclusion and exclusion criteria of the study. The subjects were recruited from several departments in the Cancer Center of Zhejiang Provincial People's Hospital between August 2022 and May 2023, including 136 males and 148 females, aged 37–85 years. They included 114 non-anemic and 170 anemic cancer patients. The distribution of different type of cancers was listed in Table 1. The diagnosis of anemia was made according to the criteria: Hb<130 g/L for males and <115g/L for females from the Chinese adult reference intervals for blood cell analysis (*Shang et al., 2013*). The inclusion criteria included: (1) presence of untreated malignancy; (2) presence of untreated anemia. The exclusion criteria were as follows: (1) presence of thrombotic diseases; (2) presence of primary liver and kidney disorders; (3) a recent history of surgery; (4) presence of cardiovascular and cerebrovascular diseases;

**Table 1  The distribution of malignancies in cancer patients with anemia and non-anemia.**

| Type of cancers | Anemia group (*n*) | Non-anemia group (*n*) | Pecentage of anemia (%) |
|---|---|---|---|
| Total cancers | 176 | 108 | 62.0 |
| Lung cancer | 17 | 4 | 80.9 |
| Colon malignancy | 17 | 7 | 70.8 |
| Cholangiocarcinoma | 15 | 1 | 93.8 |
| Gastric malignancy | 14 | 6 | 70.0 |
| Liver malignancy | 12 | 3 | 80.0 |
| Rectal malignancy | 10 | 12 | 45.5 |
| Thyroid malignancy | 8 | 11 | 42.1 |
| Prostate malignancy | 8 | 9 | 47.1 |
| Renal cancer | 8 | 5 | 61.5 |
| Pancreatic malignancy | 8 | 5 | 61.5 |
| Breast malignancy | 7 | 5 | 58.3 |
| Endometria malignancy | 7 | 4 | 63.6 |
| Cervical malignancy | 5 | 2 | 71.4 |
| Osteocarcinoma | 5 | 5 | 50.0 |
| Brain malignancy | 5 | 10 | 33.3 |
| Bladder malignancy | 5 | 3 | 62.5 |
| Esophagus malignancy | 5 | 4 | 55.6 |
| Retroperitoneal malignancy | 4 | 4 | 50.0 |
| Nasopharyngeal carcinoma | 2 | 3 | 40.0 |
| Ovary malignancy | 2 | 2 | 50.0 |
| Lumbar malignancy | 1 | 0 | 100.0 |
| Intestinal malignancy | 5 | 0 | 100.0 |
| Ureteral malignancy | 6 | 0 | 100.0 |
| Parotid gland malignancy | 0 | 3 | 0 |

(5) presence of acute infection; (6) diseases of hematological and hematopoietic system; (7) malignancies complicating hemorrhage such as colon, gastric, rectal, renal, ureteral, cervica, bladder, and nasopharyngeal malignancies. In this study, bone marrow puncture, blood smear observation, flow cytometry analysis, and hemolysis tests were used to exclude the hematological and hematopoietic malignancies, and primary hemorrhagic diseases were screened by using platelet count and blood coagulation tests. Written informed consent was obtained from all individual participants included in the study. This study was performed in line with the principles of the Declaration of Helsinki. Approval was granted by the ethical committee of Zhejiang Provincial People's Hospital (approval No. 2022KT068).

## Laboratory assays

Blood samples were collected and treated as previously described in Huang QH (*Huang et al., 2023*). In brief, peripheral blood of the patients were collected in vaccutainer tubes with EDTA-K2, sodium citrate, and anticoagulants-free (BD, Franklin Lakes, NJ, USA),

respectively, after an overnight fast before treatment. The samples with EDTA-K2 were analyzed by using the Mindray BC-7500 blood analyzer (Mindray Inc., Shenzhen, China), and the observed indice were as follows: white blood cells (WBC), RBC, platelet and reticulocyte count, RBC and reticulocyte indices (hemocrit, HCT; mean corpuscular volume, MCV; mean corpuscular hemoglobin, MCH; mean corpuscular hemoglobin concentration, MCHC; red cell distribution width-coefficient of variation, RDW-CV), immature reticulocyte indices [high, medium, and low fluorescence reticulocyte (HFR, MFR and LFR); immature reticulocyte fraction (IRF), reticulocyte production index (RPI), reticulocyte hemoglobin content (RHE)], and reticulocyte size index (mean reticulocyte volume, MRV). Subsequently, the blood samples with sodium citrate and without anticoagulants were centrifuged at $1,500 \times g$ at room temperature for 5 and 10 min to isolate sera and plasma for further measurements, respectively. Subsequently, the levels of serum albumin (ALB), creatinine (sCre), cholesterol (CHOL), and thyroid stimulating hormone (TSH), and plasma fibrinogen (Fbg) were measured by an AU5800 biochemistry analyzer (Beckman-coulter, USA) and CN-9000 coagulation analyzer (Sysmex, Kobe, Japan), respectively.

## Statistical analysis

In this study, the included subjects were far more than that of the minimal sample size (including the 20% possible missing samples), thus some missing data did not influence the statistics. The distribution normality of the data was first analyzed by the Kolmogorov–Smirnov test, and data for normal and non-normal distribution were presented as $\overline{x} \pm sd$ and median, respectively. Student's $t$-test and Mann–Whitney $U$ test were, respectively, used to analyze the samples of normality and non-normality distribution. Categorical data (percentage) were analyzed by Chi-square test. The receiver operating characteristic (ROC) curves were established to calculate the area under the curve (AUC), and the identifying ability (sensitivity and specificity) of MRV between non-anemia and anemia in cancer patients was evaluated. An adjusted-multivatiate analysis was performed by using the potential risk factors to calculate the odds ratios (OR) and 95% confidence intervals (95% CI) of MRV for early CRA. Pearson analysis was used to observe the correlation of MRV with RBC and reticulocyte indices. The SPSS 20.0 software (SPSS, IBM Corp., Armonk, NY, USA) was used, and $p$-values of less than 0.05 were considered statistically significant.

## RESULTS

### Basic physiological and laboratory characteristics of cancer patients

The study initially investigated the basic physiological and laboratory characteristics of the patients with anemia and non-anemia, respectively. A total of 23 and 21 types of tumors were included in the anemia and the non-anemia group, respectively. Although the the composition ratio of different malignancies differed between the anemia and non-anemia group, the total percentage of the main malignancies (lung cancer, gastrointestinal malignancy, and ovarian cancers) most frequently associated with anemia showed no statistical difference between anemia and non-anemia patients (65/176, 36.9% $vs.$ 31/108, 28.7%, $p > 0.05$) (Table 1). The mean age of non-anemia patients at diagnosis had a lower

years old than that of anemia one ($54.5 \pm 15.9$ *vs.* $62.1 \pm 15.7$, $p < 0.001$), and male and female patients constituted approximately the same proportion in the non-anemia and anemia groups (males: 49% *vs.* 54%, $p > 0.05$). There was no significant difference in MCV, RHE and MRV levels from reticulocyte analyse between the two groups ($p > 0.05$), but the levels of reticulocyte count and other RBC indices were significantly different ($p < 0.05$ or <0.001), which was similar to the expected results. The detailed comparisons of the basic biological and laboratory characteristics are presented in Table 2.

## Associations of MRV levels with the RBC and reticulocyte indices

The Pearson correlation analysis revealed that there was a negative correlation of MRV levels with that of RBC and RDW-CV ($r = -0.291$ and $-0.327$, and $-0.128$ and $-0.387$, respectively, both $p < 0.01$) in patients with anemia and non-anemia. A complete correlation was found between RHE and MRV levels ($r = 1.000$, $p < 0.001$), and MRV was also highly correlated with the levels of MCH, MCV and MCHC in both anemia patients ($R$: 0.575–0.807) and non-anemia ones ($R$: 0.638–0.820), respectively (Table 3).

## ROC curves analyse of MRV in identifying CRA diagnosed by RHE and RPI

The patients were first classified as anemia and non-anemia ones by the combination of RHE and RPI levels. The classification criteria for anemia were based on the single or simultaneous decrease in MRV and RHE levels compared with their median levels. All other indices of RBC and reticulocyte were treated as the independent variables. ROC curves analyse indicated a high area under curve of 0.829 (95% CI [0.762–0.895]) and 0.884 (95% CI [0.831–0.936]) for MRV in identifying anemia in male and female patients, respectively ($p < 0.001$). Each other index had a lower AUC than MRV (ranged from 0.384 to 0.795 in males and 0.343 to 0.811 in females), (Table 4). When the optimal cutoff values of MRV were set at 100.95 fl in males and 98.35 fl in females, the sensitivity and specificity were 1.00 and 0.68, and 1.00 and 0.73, respectively (Fig. 1, Table 4).

## RBC and reticulocyte indices levels in anemia and non-anemia patients classified by MCHC, MCH and MCV

Because of the highly close correlation of MRV with MCH, MCHC and MCV, the more sensitive indices of overt anemia than only Hb concentrations in clinical practice, the three ones were used to classify anemia in this study. The classification criteria for anemia were also according to the single or simultaneous decrease in MCH, MCHC and MCV levels compared with their optimal cutoff values from the ROC curves analyse in identifying CRA diagnosed by RHE and RPI. Under this type of anemia-classifying, no statistical differences were found in age and gender percent between anemia and non-anemia patients ($p > 0.05$), but the MRV, RPI, and RHE levels exhibited significant differences ($p < 0.001$) (Table 5).

## Adjusted multivariate analysis of MRV and Hb for risk of CRA

Based on the anemia classified by the combination of MCV, MCH and MCHC, some confounders such as gender, age, BMI, and blood pressure that could potentially affect the production and release of reticulocyte were included to perform an adjusted multivariate

**Table 2  Comparisons of basic characteristics and laboratory indices between anemic and nonanaemic patients with cancer.**

| Indices | Cancer without anemia | Cancer with anemia | Statistical value | *p*-value |
|---|---|---|---|---|
| n | 114 | 170 | | |
| Male(n, %) | 50 (43.9) | 86 (50.6) | 2.412 | 0.120 |
| Age (years) | 54.5 ± 15.9 | 62.1 ± 15.7 | 3.977 | <0.001 |
| BMI (Kg/m2) | 23.59 ± 3.45 | 21.91 ± 3.60 | 1.132 | 0.259 |
| SBP (mmHg) | 126.4 ± 19.3 | 128.4 ± 18.7 | 0.801 | 0.424 |
| DBP (mmHg) | 76.4 ± 10.4 | 74.3 ± 12.5 | 1.19 | 0.235 |
| sCre (mmol/L) | 65.2 (530-75.9) | 64.1 (56.0–77.5) | 0.622 | 0.534 |
| CHOL (mmol/L) | 4.73 ± 0.91 | 4.23 ± 1.16 | 0.380 | 0.705 |
| ALB (g/L) | 40.61 ± 4.10 | 35.01 ± 5.79 | 4.498 | <0.001 |
| Fbg (g/L) | 2.96 ± 0.77 | 3.52 ± 1.25 | 0.811 | 0.418 |
| TSH (U/L) | 1.69 (0.91–3.55) | 2.04 (1.36–2.84) | 1.385 | 0.166 |
| WBC ($\times 10^9$ /L) | 5.92 (4.66–7.16) | 5.62 (4.50–6.79) | 1.894 | 0.058 |
| RBC ($\times 10^{12}$ /L) | 4.17 (3.71–4.53) | 3.72 (3.35–4.17) | 11.493 | <0.001 |
| HGB (g/L) | 128.5 (115.3–142.8) | 111.0 (101.0–125.0) | 13.259 | <0.001 |
| HCT (%) | 37.6 (33.6–41.5) | 33.1 (30.0–36.5) | 13.017 | <0.001 |
| MCV (fL) | 90.1 (87.2–93.0) | 90.4 (84.2–93.4) | 1.936 | 0.053 |
| MCH (pg) | 31.0 ± 1.8 | 29.5 ± 4.1 | 3.846 | <0.001 |
| MCHC (g/L) | 342.3 ± 8.6 | 334.2 ± 15.3 | 2.362 | 0.019 |
| RDW-CV (%) | 13. 3(12.8–14.1) | 13.8 (13.0–15.2) | 6.115 | <0.001 |
| PLT ($\times 10^9$ /L) | 216 (176–266) | 205 (158–274) | 0.002 | 0.998 |
| Ret# ($\times 10^9$ /L) | 73 (54–90) | 64 (52–81) | 2.161 | 0.031 |
| HFR (%) | 0.1 (0.0–0.9) | 0.8 (0.0–2.3) | 7.994 | <0.001 |
| MFR (%) | 7.77 ± 2.47 | 11.07 ± 4.24 | 3.319 | 0.001 |
| LFR (%) | 91.92 ± 2.75 | 86.71 ± 6.91 | 3.498 | 0.001 |
| IRF (%) | 8.08 ± 2.75 | 13.30 ± 6.91 | 3.498 | 0.001 |
| RPI | 1.1 (0.7–1.5) | 0.8 (0.6–1.1) | 7.293 | <0.001 |
| RHE (pg) | 28.6 (27.5–29.7) | 28.2 (25.6–29.9) | 1.949 | 0.051 |
| MRV (fL) | 100.6 (96.7–104.2) | 99.2 (90.2–104.9) | 1.862 | 0.063 |

Notes.

Data were presented as mean ± standard deviation, median (P25–P75) or percentage.

BMI, body mass index; SBP, systolic blood pressure; DBP, diastolic blood pressure; WBC, white blood cells; RBC, red blood cell; PLT, platelet; HCT, hematocrit; MCV, mean corpuscular volume; MCH, mean corpuscular hemoglobin; MCHC, mean corpuscular hemoglobin concentration; RDW-CV, red cell distribution width-coefficient of variation; RET, reticulocyte count; HFR, high fluorescence reticulocyte; MFR, medium fluorescence reticulocyte; LFR, low fluorescence reticulocyte; IRF, immature reticulocyte fraction; RPI, reticulocyte production index; RHE, reticulocyte hemoglobin content; MRV, mean reticulocyte volume; alb, albumin; Cre, creatinine; CHOL, cholesterol; TSH, thyroid stimulating hormone; Fbg, fibrinogen..

*P* values were calculated by student's *t*-test, Mann-Whitney *U*-test and Chi-square test, respectively.

regression analysis. The logistic regression analyse showed that both MRV and Hb are the independent risk factors for anemia, and as the continuous variables, the risk of anemia increased by almost 0.1 and 0.04 times for each 1 fl and 1g reduction of MRV and Hb levels, respectively. When MRV and Hb were treated as categorical variables expressed as "increased" or "decreased" by using the optimal cutoff values of male and female patients

**Table 3  Correlation of MRV levels with the RBC and reticulocyte indices in cancer patients.**

| Grouping | r/p | RHE | RPI | RET# | RBC | HGB | MCV | MCH | MCHC | RDW-CV |
|---|---|---|---|---|---|---|---|---|---|---|
| Anemic patients | r | 1.000 | 0.318 | 0.091 | −0.291 | 0.261 | 0.775 | 0.807 | 0.602 | −0.337 |
| | p | <0.001 | <0.001 | >0.05 | <0.01 | <0.001 | <0.001 | <0.001 | <0.001 | <0.001 |
| Non-anemic patients | r | 1.000 | 0.415 | 0.089 | −0.128 | 0.507 | 0.796 | 0.820 | 0.566 | −0.387 |
| | p | <0.001 | <0.001 | >0.05 | <0.001 | <0.001 | <0.001 | <0.001 | <0.001 | <0.001 |

Notes.

RBC, red blood cell; MCV, mean corpuscular volume; MCH, mean corpuscular hemoglobin; MCHC, mean corpuscular hemoglobin concentration; RDW-CV, red cell distribution widthcoefficient of variation; RET#, reticulocyte count; RPI, reticulocyte production index; RHE, reticulocyte hemoglobin content; MRV, mean reticulocyte volume.

*P* values were calculated by Spearman analysis.

**Table 4  ROC curve analysis of MRV in identifying anemia diagnosed by the combination of RHE and RPI.**

| Variables | AUC | 95%CI | | *p*-value | Optimal cutoff values | Sensitivity | Specificity | Youden index |
|---|---|---|---|---|---|---|---|---|
| | | Lower | Upper | | | | | |
| **Males** | | | | | | | | |
| MRV | 0.829 | 0.762 | 0.895 | <0.001 | 100.95 | 1.00 | 0.68 | 0.68 |
| MCV | 0.795 | 0.721 | 0.869 | <0.001 | 90.65 | 0.90 | 0.59 | 0.49 |
| MCH | 0.782 | 0.700 | 0.865 | <0.001 | 31.85 | 0.65 | 0.79 | 0.44 |
| RET# | 0.782 | 0.704 | 0.859 | <0.001 | 0.07 | 0.88 | 0.59 | 0.47 |
| HGB | 0.726 | 0.640 | 0.812 | <0.001 | 109.50 | 0.95 | 0.40 | 0.35 |
| HCT | 0.716 | 0.627 | 0.804 | <0.001 | 33.80 | 0.88 | 0.49 | 0.36 |
| RBC | 0.627 | 0.531 | 0.724 | 0.019 | 3.68 | 0.85 | 0.43 | 0.28 |
| MCHC | 0.594 | 0.489 | 0.700 | 0.084 | 347.50 | 0.38 | 0.80 | 0.18 |
| RDW-CV | 0.384 | 0.284 | 0.485 | 0.034 | 12.10 | 1.00 | 0.02 | 0.02 |
| **Females** | | | | | | | | |
| MRV | 0.884 | 0.831 | 0.936 | <0.001 | 98.35 | 1.00 | 0.73 | 0.73 |
| MCH | 0.811 | 0.742 | 0.880 | <0.001 | 30.75 | 0.77 | 0.76 | 0.53 |
| RET# | 0.786 | 0.714 | 0.859 | <0.001 | 0.06 | 0.96 | 0.50 | 0.46 |
| MCV | 0.768 | 0.693 | 0.842 | <0.001 | 85.75 | 0.98 | 0.44 | 0.42 |
| MCHC | 0.721 | 0.636 | 0.805 | <0.001 | 341.50 | 0.65 | 0.72 | 0.37 |
| HGB | 0.697 | 0.612 | 0.782 | <0.001 | 109.50 | 0.79 | 0.53 | 0.32 |
| HCT | 0.668 | 0.578 | 0.757 | 0.001 | 32.65 | 0.79 | 0.52 | 0.31 |
| RBC | 0.498 | 0.401 | 0.595 | 0.971 | 2.89 | 1.00 | 0.07 | 0.07 |
| RDW-CV | 0.343 | 0.255 | 0.431 | 0.002 | 12.15 | 0.98 | 0.06 | 0.03 |

Notes.

MRV, mean reticulocyte volume; RPI, reticulocyte production index; RHE, reticulocyte hemoglobin content; RBC, red blood cell; HCT, hematocrit; MCV, mean corpuscular volume; MCH, mean corpuscular hemoglobin; MCHC, mean corpuscular hemoglobin concentration; RDW-CV, red cell distribution width-coefficient of variation; RET#, eticulocyte count; ROC, receiver operating characteristic; AUC, area under curve; CI, confidence interval.

*P* values were calculated by ROC curve analysis.

from ROC curve analyse, the overall risk of anemia in patients with below the cutoff values was 19 and 4 times higher than that with above the cutoff values, respectively (Table 6). Moreover, when all cancer patients were divided into four groups by the quartiles of MRV

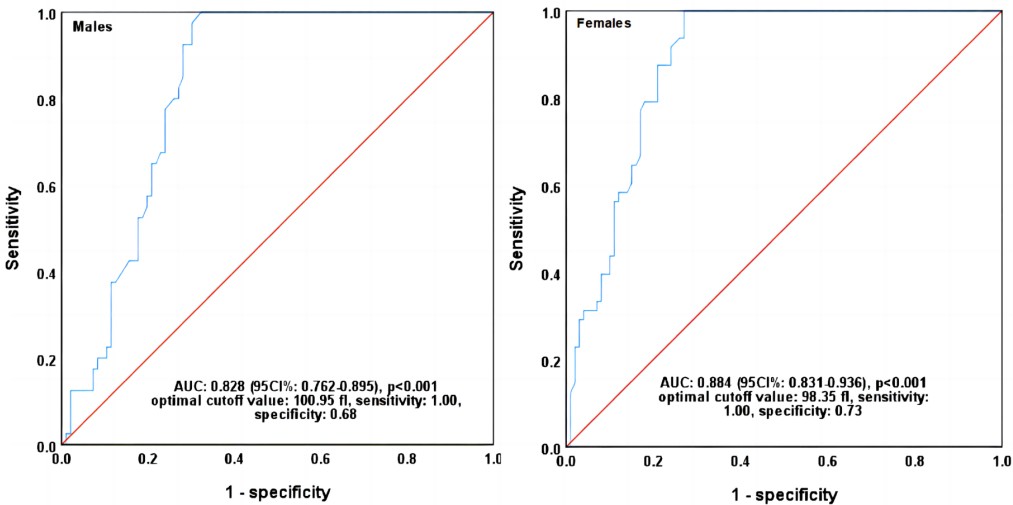

**Figure 1** **ROC curves of MRV in identifying CRA diagnosed by RHE and RPI.** MRV, mean reticulocyte volume; CRA, cancer-related anemia; RPI, reticulocyte production index; RHE, reticulocyte hemoglobin content; ROC, receiver operating characteristic; AUC, area under curve; CI, confidence interval. *P* values were calculated by ROC curve analysis.

**Table 5** **Comparisons of the various indicators levels in anemia and non-anemia grouped by MCV, MCH and MCHC.**

| Grouping | n | Male (n, %) | Age (year) | MRV (fl) | RHE (pg) | RPI | RET# (x $10^{12}$/L) | MCV (fl) |
|---|---|---|---|---|---|---|---|---|
| Non-anemia | 61 | 20 (33.3) | 58.0 (44.5–71.5) | 104.55 ± 6.28 | 29.84 ± 1.89 | 1.1 (0.8–1.5) | 0.074 (0.054–0.092) | 92.5 (91.1–95.0) |
| Anemia | 223 | 116 (52.0) | 62.0 (47.0–72.0) | 96.99 ± 12.73 | 27.60 ± 3.81 | 0.80 (0.60–1.20) | 0.065 (0.053–0.083) | 89.0(83.7-92.4) |
| Statistical value | | 7.098 | 0.733 | 4.487 | 4.433 | 3.172 | 1.736 | 6.115 |
| *P*-value | | 0.008 | 0.464 | <0.001 | <0.001 | 0.002 | 0.082 | <0.001 |

**Notes.**

Data were presented as mean ± standard deviation, median (P25–P75) or percentage.

MRV, mean reticulocyte volume; RHE, reticulocyte hemoglobin content; RPI, reticulocyte production index; MCV, mean corpuscular volume; MCH, mean corpuscular hemoglobin; MCHC, mean corpuscular hemoglobin concentration; RET#, reticulocyte count.

*P* values were calculated by student's *t*-test, Mann–Whitney *U*-test and Chi-square test, respectively.

and Hb levels of male and female patients, the incidence of overall anemia demonstrated an increase trend along with the decrease of MRV and Hb quartiles, respectively (p-trend<0.01), but which was based on MRV quartile showed a more significant decrease (Fig. 2).

## DISCUSSION

In the present study, besides the predictable abnormalities of the basic indices of RBC and reticulocyte in anemia patients, we found that the RHE and MRV levels did not exhibit marked differences between the two groups. The results indicated the level changes of these reticulocyte indices were inconsistent with the decrease of RBC indices levels in

**Table 6  Multivariate regression analysis of MRV for CRA diagnosed by MCV, MCH and MCHC.**

| Variables | OR (95%CI) | *p*-value |
|---|---|---|
| MRV | | |
| Continuous variable | 1.096 (1.055–1.138) | <0.001 |
| Optimal cutoff from ROC curve | | |
| male>100.95fl, female>98.35fl | 1 | |
| male<100.95fl, female<98.35 | 19.111 (6.985–52.288) | <0.001 |
| Hb | | |
| Continuous variable | 1.037 (1.018–1.057) | <0.001 |
| Optimal cutoff from ROC curve | | |
| male>109.5g/L, female>109.5g/L | 1 | |
| male<109.5g/L, female<109.5g/L | 3.864 (1.763–8.467) | <0.01 |

**Notes.**

The results were analyzed by adjusting risk factors of sex, age, BMI, SBP, DBP when the indices were presented as continuous and categorical variables based on the cut-off values, respectively.

CRA, cancer-related anemia; BMI, body mass index; SBP, systolic blood pressure; DBP, diastolic blood pressure; MRV, mean reticulocyte volume; Hb, hemoglobin; MCV, mean corpuscular volume; MCH, mean corpuscular hemoglobin; MCHC, mean corpuscular hemoglobin concentration; OR, odds ratio; CI, confidence interval.

*P* values were calculated by multivariate regression analysis.

anemia patients, suggesting that the early diagnosis of anemia based on hemoglobin level has some deficiency with less diagnostic power. Reticulocyte hemoglobin content has been reported as an early marker of iron-deficient erythropoiesis, and is a better predictive index than hemocytometric parameters such as MCV, RDW, and Hb (*Chinudomwong et al., 2020*). A low RHE value might be an early marker of iron deficient erythropoiesis (*Joosten et al., 2013*). Therefore, compared with reticulocyte indices, Hb is not a sensitive index in anemia diagnosis, thus some cancer patients may be with normal Hb and decreased reticulocyte generation reflected by the decreases of RHE and MRV levels, which may cause the similar overall levels of RHE and MRV between anemia and non-anemia patients classified by Hb. Therefore, some early anemia may be missed. Some studies have revealed that the RHE and RBC size factor calculated by $\sqrt{MCV*MRV}$ were sensitive indices in the diagnosis of inefficient erythropoiesis, almost the eaqual diagnostic power for iron deficient erythropoiesis (*Urrechaga, Borque & Escanero, 2011*. Thus, as a macroscopic feature of laboratory analysis for anemia diagnosis, Hb concentration is not conducive for early diagnosis of CRA.

Conventionally, iron deficiency diagnosis is based on determining ferritin, transferrin saturation, and soluble transferrin receptor, but these biomarkers are strongly influenced by inflammation and infection (*Ringoringo et al., 2023*). RHE and RPI have been recognized the sensitive indices in the diagnosis of anemia. Many studies have shown that RHE can show iron deficiency without anemia (*Ennis et al., 2018*; *Pomrop et al., 2022*; *Düzenli Kar & Altınkaynak, 2021*), and RPI can monitor bone marrow response to iron-treated anemia (*Adane & Asrie, 2021*), and it can also be used as a diagnostic guide to the etiology of anemia (*Riley et al., 2001*). Thus, combining RHE and RPI will be a better diagnostic index before IDA occurs. In this study, when the cancer patients were classified as anemia and non-anemia ones by the combination of RHE and RPI levels, the MRV showed a
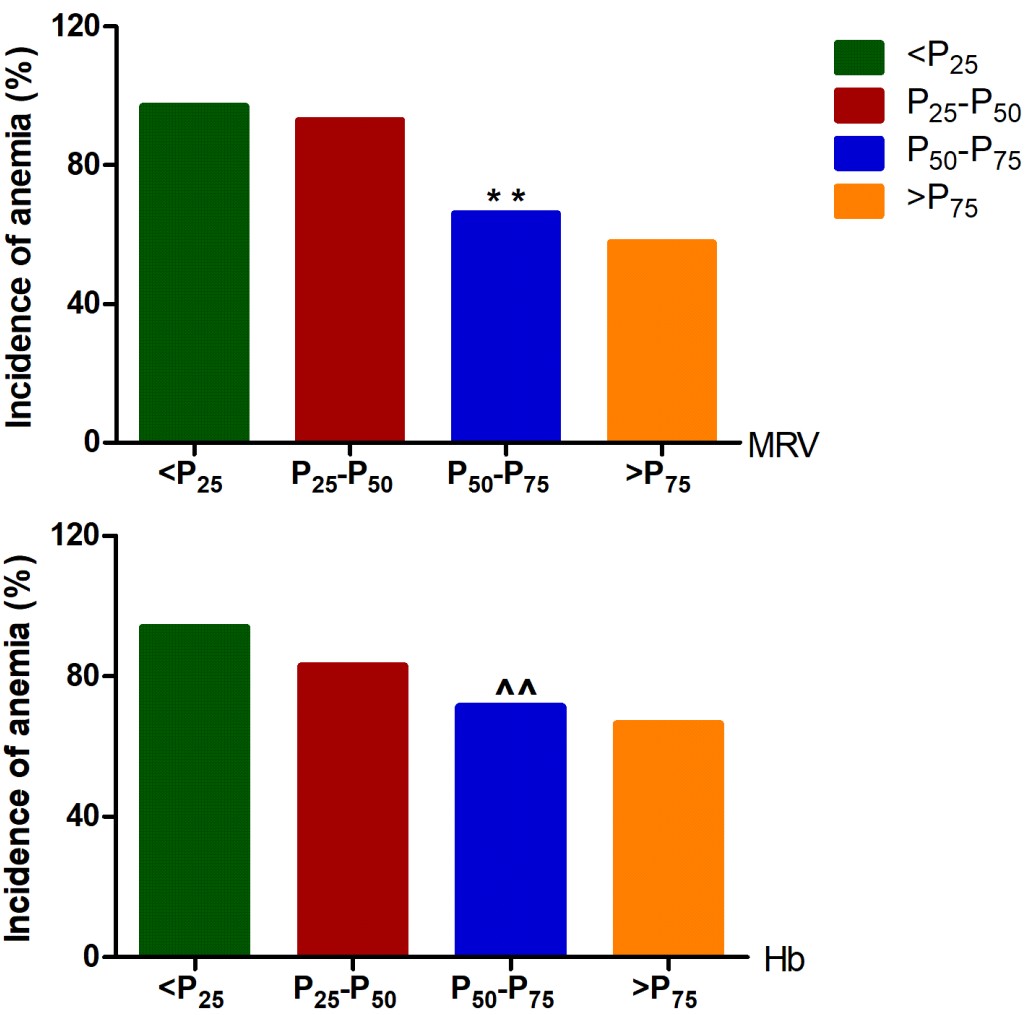

**Figure 2  The incidence of overt cancer-related anemia in different quartile of MRV levels.** In males, the $P_{25}$, $P_{50}$, and $P_{75}$ of MRV were 95.38, 101.05, and 106.50fl, and which of Hb was 104.3, 122.0, and 141.75 g/L, respectively. In females, the $P_{25}$, $P_{50}$, and $P_{75}$ of MRV were 92.43, 98.55, and 103.45fl, and which of Hb was 105.0, 112.0, and 125.0 g/L, respectively. MRV: mean reticulocyte volume; Hb: hemoglobin. Among the different qutile, the trend- $p < 0.001$ analyzed by trend Chi-square test; \*\* $p < 0.001$, and ^^ $p > 0.05$ compared with $P_{25}$-$P_{50}$ analyzed by Chi-square test, respectively.

much higher area under ROC curve than that of any other index in identifying anemia and non-anemia of either male or female patients. Furthermore, when the optimal cutoff values were set at 100.95 fL for males and 98.35fL for females, the sensitivity of MRV in both was found at 100% with the specificity of 63% and 76%, respectively. All the cumulative evidence indicates that MRV is a much more sensitive index than the parameters of RBC in the diagnosis of anemia, and it can be used as a powerful diagnostic parameter of early anemia in cancer patients.

At present, Hb, MCV, MCH, and MCHC are the most commonly used hematological screening indices, but they are derived from the mature RBC, with a life span of about 120 days, and therefore takes a long time to reflect the the state of iron deficiency. Consequently,

anemia screening by one of the mature RBC indices will delay the detection of inefficient erythropoiesis. MRV represents the size or volume of the more recently produced RBCs, within a period of <3 days before the measurement (*Urrechaga, Borque & Escanero, 2011*. Lower reticulocyte volume will finally lead to a low RBC volume, which is presented as an early low MRV and subsequent low MCV. In the study, MRV exhibited a correlation coefficient of 1.000 with RHE in both anemia and non-anemia patients, suggesting the almost similar power in the diagnosis of anemia for the two indices. Although there was a high correlation of MRV with MCV, MCH, and MCHC levels, the correlation was lower between MRV and RBC and Hb, respectively, which further implies Hb is not conductive for early diagnosis of CRA, and MRV is a better index.

MCV, MCH and MCHC can reflect the size of RBC and Hb content and concentration in single RBC, respectively. In this study, we used a parallel trial approach by combining MCV, MCH and MCHC to divide the patients to anemic and non-anemic ones. A marked difference of MRV level was found between the two groups, and MRV level significantly decreased in anemic patients, further indicating a less sensitivity of Hb concentration in the early diagnosis of overt CRA. As representing the sensitive laboratory indices of anemia, the combining use of MCV, MCH and MCHC is much better than Hb in the diagnosis of overt anemia. It has been recognized that all mature RBC indices are the embodiment of the laboratory analysis of anemia based on hemoglobin level, which is the apparent indices of clinical diagnosis and the final result of the pathological process of anemia development. Thus, monitoring the evolution of the pathological process is more important to give early prediction of anemia. When CRA was classified by the above combination, an adjusted multivariate regression analysis revealed that either MRV or Hb is the independent risk factor for anemia with a high OR value, respectively. However, a much higher OR of 19.1 for MRV was found, which suggesting MRV may be a more powerful predicting index of overt anemia than Hb concentration in cancer patients. Consistent with these results, we also observed an decreased incidence of CRA with the increment of MRV level and Hb concentration, but a more significant decrease of the incidence of CRA was closely correlated with the increase of MRV level. As a convenient and fast calculated index on Mindray BC-7500 analyzer, MRV demonstrated its special clinical value in the diagnosis and risk evaluation of CRA. In recent years, few studies have evaluated the significance of MRV in CRA because it is an unavailable index on most analyzers and is not commonly used to predict and assess risk in overt CRA. Similarly, the present study suggests that the MRV is an powerful and practical index for the early diagnosis of CRA.

It has been recognized that cancer progression and treatment can affect RBC size, and the MCV is a good marker of treatment response and prognosis in cancers (*Zhang et al., 2018*; *Li et al., 2020*; *Wang & Zhang, 2022*). Both cancer progression and treatment can lead to anemia by inhibiting bone marrow hematopoiesis, and they can also cause the abnormalities of reticulocyte size, thus significantly affect the treatment response and prognosis. Therefore, besides as a diagnostic index, MPV may also help in the assessment of treatment effect and prognosis of CRA patients, which neesds in-depth observation to reveal and clarify the more valuable usefulness of MRV than other indice of mature RBC and reticulocyte in CRA. On the other hand, our results were obtained from the

Mindray BC-7500 analyzer, and we do not know whether the measurements of MRV and other indice on other blood analyzers can produce similar results in the early diagnosis, identification and prediction of CRA, which suggests that it would be of importance to draw a definite conclusion for the use of different blood analyzers. Moreover, whether MRV is a valuable index in chronic diseases seems to be unclear, and the further exploration in chronic diseases will reveal more clinical significance of MRV and extend its use scope in the early diagnosis of all chronic diseases-related anemia.

Within our knowledge, there are some limitations in this study. Firstly, CRA is the functional iron deficiency-related anemia, and is not true iron deficiency. In this study, the diagnosis of anemia was made according to the reference interval of peripheral hemoglobin of Chinese adult males and females, and we did not included the serum iron and ferritin to identify the iron deficiency and non-iron deficiency of patients, thus to evaluate the diagnostic power of MRV for CRA. Therefore, the actual significance of MRV in the diagnosis of CRA has not been explored and fully elucidated. Secondly, since the cancer patients of this study included multiple cancer types, and the baseline levels of RHE and MRV in the different cancer patients might not be comparable, thus it may lead to the non-significant differences between the results of the two groups. Therefore, inclusion of the same type of cancer population with appropriate sample size for further study may draw more reliable conclusions. Despite the limitations our study also revealed that decreased MRV level was strongly correlated with early anemia in cancer patients.

## CONCLUSION

This study revealed that the MRV can be used as a sensitive index in early diagnosis of CRA, and decreased MRV level may be the powerful risk factor of overt anemia in cancer patients. However, further studies from multi-center and different analyzers with more diverse patients with CRA are needed to draw more definitive conclusions.

### Funding

This work was supported by the Medicine and Health Science and Technology Project of Zhejiang Province, China (No. 2024KY741 to Huijun Lin, No. 2024KY765 and 2021KY060 to Xianming Fei), and the Tradictional Chinese Medicine Science and Technology Project of Zhejiang Province, China (No. 2024ZL253) to Yan Yu. The funders had no role in study design, data collection and analysis, decision to publish, or preparation of the manuscript.

### Grant Disclosures

The following grant information was disclosed by the authors:
Medicine and Health Science and Technology Project of Zhejiang Province, China: 2024KY741, 2024KY765, 2021KY060.
Tradictional Chinese Medicine Science and Technology Project of Zhejiang Province, China: 2024ZL253.

## Competing Interests

The authors declare there are no competing interests.

## Author Contributions

- Huijun Lin conceived and designed the experiments, performed the experiments, analyzed the data, prepared figures and/or tables, authored or reviewed drafts of the article, and approved the final draft.
- Bicui Zhan performed the experiments, analyzed the data, prepared figures and/or tables, authored or reviewed drafts of the article, and approved the final draft.
- Xiaoyan Shi performed the experiments, prepared figures and/or tables, authored or reviewed drafts of the article, and approved the final draft.
- Dujin Feng performed the experiments, analyzed the data, authored or reviewed drafts of the article, and approved the final draft.
- Shuting Tao performed the experiments, authored or reviewed drafts of the article, and approved the final draft.
- Mingyi Wo performed the experiments, analyzed the data, authored or reviewed drafts of the article, and approved the final draft.
- Xianming Fei conceived and designed the experiments, analyzed the data, prepared figures and/or tables, authored or reviewed drafts of the article, funding acquisition, and approved the final draft.
- Weizhong Wang conceived and designed the experiments, authored or reviewed drafts of the article, project administration, and approved the final draft.
- Yan Yu conceived and designed the experiments, analyzed the data, authored or reviewed drafts of the article, project administration, and approved the final draft.

## Human Ethics

The following information was supplied relating to ethical approvals (i.e., approving body and any reference numbers):

The ethical committee of Zhejiang Provincial People's Hospital

## Data Availability

The raw measurements are available in the Supplementary File.

## Supplemental Information

Supplemental information for this article can be found online at http://dx.doi.org/10.7717/peerj.17063#supplemental-information.

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
