# Peer review of "The mean reticulocyte volume is a valuable index in early diagnosis of cancer-related anemia"

_PeerJ, doi:10.7717/peerj.17063_

## Round 0.1 · original submission · Major Revisions

The authors should address the issues raised by the reviewers. Also, please consider whether you may omit the kind of blood analyser from the title (this might not be a peculiarity of the analyser mentioned by the authors), and whether you have elements to extend your observations to chronic, but not oncologic diseases. Please discuss.

Reviewer 1 ·

Basic reporting

This study aimed to assess the potential of mean reticulocyte volume (MRV) obtained from Mindray BC-7500 analyzer in early prediction of Cancer-Related Anemia (CRA). They found that this MRV was the best predictor in this situation.

The study was well designed and the manuscript was well written. However, some points are needed to be revised or corrected before final decision, including

1. Some abbreviations seemed not relevant with the full term such as CRA which should not abbreviate Cancer Associated Anemia. RHE also should not abbreviate Reticulocyte Hemoglobin Content. Please consider these abbreviations and make a change.

2. As Hb cannot be used to classify anemia and non-anemia in this cancer patients analyzed, combinations of either reticulocyte indices or RBC indices were used for classifying anemia and non-anemia. However, detail of this means of classification was missing. Please also add this information in the content to make this manuscript clear.

Experimental design

.

Validity of the findings

.

Annotated reviews are not available for download in order to protect the identity of reviewers who chose to remain anonymous.

·

Basic reporting

In this study, the early diagnosis of anemia in the context of neoplastic diseases is discussed. Specifically, the reduction of the mean reticulocyte volume (MRV), measured on the Mindray BC-7500 analyzer, is explored as a potential tool in detecting the early onset of anemia due to functional iron deficiency (IDA). The manuscript maintains consistency throughout the text and demonstrates a satisfactory use of the English language. However, specific aspects need to be revised or corrected before a final decision can be reached. Addressing these points will enhance the overall quality and coherence of the study, making it more suitable for publication.


1. At the line 224 acronyms SBP and DBP should be written extensively. Please review these abbreviations and make any necessary adjustments.

2. As highlighted in lines 252-253, the RSf index serves as a sensitive indicator of inefficient erythropoiesis, and, as described in the literature, it shares the same clinical significance of Ret He. How is MRV more advantageous for the early diagnosis of anemia? Since its specificity is low, how would it be beneficial for this kind of patients? Please provide additional details regarding the clinical advantages, such as its role in follow-up or early therapy.

Experimental design

The methodology is clearly explained and well-constructed; however, a revision of the study’s design definition could enhance comprehension of the process.

1) At line 127, the study is defined as 'Prospective.' Is this term sufficient to elucidate the study's design, or could it be also defined as cross-sectional? Please revise this information to enhance clarity in the manuscript.

2)The description of sample processing requires greater clarity, as from lines 146-147 it seems that all samples are centrifuged, while it is not specified that not centrifuged blood in K-EDTA containing tubes is used for the analysis of hematological parameters. In order to avoid misunderstanding for readers who are not specialists in the field, it would be beneficial to reorganize the description of the phases related to sample processing.

Validity of the findings

The study aims to demonstrate that the MRV measured on the Mindray BC-7500 can serve as a convenient and sensitive indicator for the early diagnosis of cancer-related anemia. Nevertheless, a broader point of view could be helpful for others who may wish to replicate the experiment. Expanding the discussion by comparing results with those obtained by others with other blood count instruments could enhance (and possibly generalize) the validity and strength of your findings. In particular:

1) Regarding the measurement of MRV, since it can be assessed on automatic analyzers other than the Mindray BC-7500, would the parameters be comparable across different analyzers? A referring to this matter could be beneficial.

·

Basic reporting

1.1 The manuscript structure is clear and raw data is supplied. However, there are some wordings of the manuscript that are duplicates such as line 182.
1.2 Literature references are sufficient.

Experimental design

2.1 This research was conducted on the Chinese population, which has a high frequency of other red blood cell disorders such as thalassemia, G6PD deficiency, and red blood cell membrane disorders. Therefore, laboratory investigations of differential diagnosis of these conditions should be included in the experimental design for categorizing cancer patients with or without red blood cell disorders.
2.2 Two hundred and eighty-four patients from different types of cancers (more than 20 types) were recruited for this study. 170 Anemic and 114 non-anemic patients were observed. Some types of cancer were directly associated with anemia by affecting blood cell formation or destruction of red blood cells or chronic blood loss. Therefore, some types of cancers were considered for exclusion from the analysis.
2.3 The frequency of anemia in each type of cancer is suggested for inclusion in the manuscript.

Validity of the findings

3.1 This study used many types of cancers, it is a more valid finding if the same type of cancer (with an appropriate sample size) is used for data analysis.

---

## Round 0.2 · Minor Revisions

Please address the final issues raised by reviewer 3.

Reviewer 1 ·

Basic reporting

All points met the standard.

Experimental design

All points met the standard.

Validity of the findings

All points met the standard.

Additional comments

All suggested points were corrected accordingly.

·

Basic reporting

The authors have appropriately revised the required points.

Experimental design

The authors have appropriately revised the required points.

Validity of the findings

The authors have appropriately revised the required points.

·

Basic reporting

The manuscript is clear and raw data is shared. Concerning to figure preparation of ROC curves of MRV (both male and female) instead of AUC values stated in table 4. Therefore, the selection of the best AUC value for representing in the new figure.

Experimental design

According to the exclusion of common red blood cell disorders from this study, please state the specific tests for determining those red blood cell disorders in the manuscript.

Validity of the findings

No comment

Additional comments

The authors addressed/responded clearly to all my comments and revised the manuscript according to all comments/suggestions.

---

## Round 0.3 · accepted · Accept

The authors have fully addressed all the issues raised by the reviewers and the editor.

·

Basic reporting

The authors addressed/responded clearly to all my comments and revised the manuscript according to all comments/suggestions.

Experimental design

The authors addressed/responded clearly to all my comments and revised the manuscript according to all comments/suggestions.

Validity of the findings

No comments.

Additional comments

No comments.